# A high-quality severe combined immunodeficiency (SCID) rat bioresource

**Yoshiki Miyasaka**[1], **Jinxi Wang**[1,2], **Kosuke Hattori**[1,2], **Yuko Yamauchi**[1,2], **Miho Hoshi**[2], **Kazuto Yoshimi**[1,2], **Saeko Ishida**[2], **Tomoji Mashimo**[1,2]*

**1** Institute of Experimental Animal Sciences, Graduate School of Medicine, Osaka University, Suita City, Osaka, Japan, **2** Division of Animal Genetics, Laboratory Animal Research Center, Institute of Medical Science, The University of Tokyo, Minato-ku, Tokyo, Japan

\* mashimo@ims.u-tokyo.ac.jp

**Data Availability Statement:** All relevant data are within the paper and its Supporting information files.

**Funding:** The author(s) received no specific funding for this work.

## Abstract

Immunodeficient animals are valuable models for the engraftment of exogenous tissues; they are widely used in many fields, including the creation of humanized animal models, as well as regenerative medicine and oncology. Compared with mice, laboratory rats have a larger body size and can more easily undergo transplantation of various tissues and organs. Considering the absence of high-quality resources of immunodeficient rats, we used the CRISPR/Cas9 genome editing system to knock out the interleukin-2 receptor gamma chain gene (*Il2rg*) in F344/Jcl rats—alone or together with recombination activating gene 2 (*Rag2*)—to create a high-quality bioresource that researchers can freely use: severe combined immunodeficiency (SCID) rats. We selected one founder rat with frame-shift mutations in both *Il2rg* (5-bp del) and *Rag2* ([1-bp del+2-bp ins]/[7-bp del+2-bp ins]), then conducted mating to establish a line of immunodeficient rats. The immunodeficiency phenotype was preliminarily confirmed by the presence of severe thymic hypoplasia in *Il2rg*-single knockout (sKO) and *Il2rg*/*Rag2*-double knockout (dKO) rats. Assessment of blood cell counts in peripheral blood showed that the white blood cell count was significantly decreased in sKO and dKO rats, while the red blood cell count was unaffected. The decrease in white blood cell count was mainly caused by a decrease in lymphocytes. Furthermore, analyses of lymphocyte populations via flow cytometry showed that the numbers of B cells (CD3⁻ CD45⁺) and natural killer cells (CD3⁻ CD161⁺) were markedly reduced in both knockout rats. In contrast, T cells were markedly reduced but showed slightly different results between sKO and dKO rats. Notably, our immunodeficient rats do not exhibit growth retardation or gametogenesis defects. This high-quality SCID rat resource is now managed by the National BioResource Project in Japan. Our SCID rat model has been used in various research fields, demonstrating its importance as a bioresource.

## Introduction

Immunodeficient animals are essential tools for the creation of humanized animal models through human tissue or cell xenografts. Such humanized models are widely accepted in studies of disease pathogenesis, as well as the development of therapeutic strategies and pre-clinical

**Competing interests:** The authors have declared that no competing interests exist.

tests. Thus far, the rapid development of genome engineering technologies (e.g., genome editing) has led to the creation of many immunodeficient animal models with wide-ranging applications [1, 2]. Particularly because of the ease of working with the mouse genome and embryos, immunodeficient mice have become the main focus of heterotransplantation studies; many immunodeficient mouse models have been generated, including NOD.Cg-*Prkdc^scid*/J mice [3], NOD.Cg-*Prkdc^scid Il2rg^tm1Sug*/JicTac mice [4], and NOD-*Rag1^null Il2rg^null* mice [5]. The outstanding performances of these immunodeficient mouse models have encouraged researchers to develop additional immunodeficient models using animals that are larger and more suitable for xenografts or the transplantation of bioengineering materials. The laboratory rat is another attractive experimental animal that has been widely applied in toxicology and pharmacy research studies. Compared with mice, the larger rat body size (up to 10-fold larger than the mouse body size) provides greater blood volume and facilitates surgical engraftment. Additionally, immunological properties are more similar between rats and humans than between mice and humans [6]. For these reasons, there is value in the development of humanized immunodeficient rat models, such as the patient-derived xenograft models that have been widely applied in oncology studies and in the development of novel therapies for various cancers [7–9]. Although immunodeficient rats are suitable models, there remain limited genetic modification resources and immunodeficiency resources in rats. The severe combined immunodeficiency (SCID) mutation in the *Prkdc* gene, common in many immunodeficient mouse models, causes defects in T cells and B cells [3]. In addition to this spontaneous mutation, loss of the recombination activating gene (*Rag*), which encodes a protein that mediates the V(D)J recombination essential for lymphocyte differentiation [10], has been used in previous efforts to generate multiple immunodeficient mice [11–15]. Furthermore, immunodeficient mice [16, 17] and rats [18] with natural killer (NK) cell depletion—mediated by knocking out the interleukin 2 receptor gamma chain coding gene (*Il2rg*)—have been widely used in studies that require human cell transplantation. We previously generated SCID rats by knocking out the *Prkdc* and *Il2rg* genes [19]. However, we encountered an unexpected developmental delay phenotype, which hindered the development of high-quality bioresources.

Here, we generated SCID rats by knocking out *Rag*2 and *Il2rg* in F344/Jcl rats, using an efficient genome editing protocol that we described in a previous publication [20]. Analyses of immunoglobulins and lymphocytes in the peripheral blood of these SCID rats revealed that they had a marked immunodeficiency phenotype. Furthermore, the body weight and reproductive capacity were comparable between SCID rats and wild-type (WT) rats. The presence of obvious immunodeficiency traits, lack of growth retardation, and reproductive capacity comparable to WT rats are exceptional characteristics. We provide *Il2rg/Rag* double knockout (dKO) and *Il2rg* single knockout (sKO) SCID rats with these merits as part of the National BioResource Project in Japan [21]. A sufficiently immunosuppressed internal environment and ease of breeding support the high quality of our SCID rats; since 2017, we have been providing these SCID rats to research institutions worldwide.

## Materials and methods

### Animals

F344/Jcl rats were purchased from CLEA Japan, Inc. (Tokyo, Japan). All rats were housed in an individually ventilated cage system; they received a standard diet and tap water *ad libitum*. Microbiological analyses of *Il2rg/Rag* dKO and *Il2rg* sKO-SCID rats were conducted by the Fujinomiya Technical Service Center of CLEA Japan. All animal experiments were approved by the Osaka University Animal Experiment Committee (approval number: 24-006-042).

## Electroporation of gRNA and Cas9 mRNA into rat zygotes

Cas9 mRNA was transcribed *in vitro* using a mMESSAGE mMACHINE T7 Ultra Kit (Life Technologies, Grand Island, NY, USA) from linearized plasmid (pCas9-polyA, ID #72602; www.addgene.org/CRISPR); it was purified using a MEGAClear kit (Life Technologies). Guide RNAs (gRNAs) were designed using CRISPOR software (http://crispor.tefor.net/) that predicts unique target sites throughout the rat genome. Specific CRISPR RNAs (Alt-R CRISPR-Cas9 crRNA) were purchased from Integrated DNA Technologies (USA) and assembled with a trans-activating crRNA (Alt-R CRISPR-Cas9 tracrRNA) before use, in accordance with the manufacturer's instructions. Pronuclear-stage rat embryos were collected from 8–12-week-old female rats in which superovulation had been induced by administering 150 U/kg of pregnant mare serum gonadotropin (ASKA Animal Health Co., Tokyo, Japan), followed by 75 U/kg of human chorionic gonadotropin (ASKA Animal Health Co.). After natural mating, pronuclear-stage embryos were collected from the oviducts of the female rats and cultured in a modified Krebs–Ringer bicarbonate medium or KSOM medium (ARK Resource, Kumamoto, Japan). For electroporation, 50–100 rat embryos at 3–4 h after collection were placed in a chamber with 40 µl of serum-free media (Opti-MEM, Thermo Fisher Scientific, Waltham, MA, USA) containing 400 ng/µl Cas9 mRNA and 200 ng/µl gRNA. The embryos were electroporated with a 5-mm gap electrode (CUY505P5 or CUY520P5 Nepa Gene, Chiba, Japan) in a NEPA21 Super Electroporator (Nepa Gene). The poring pulses for electroporation were: voltage, 225 V; pulse width, 2.0 ms; pulse interval, 50 ms; and number of pulses, 4. The first and second transfer pulses were: voltage, 20 V; pulse width, 50 ms; pulse interval, 50 ms; and number of pulses, 5. Rat embryos that developed to the two-cell stage after the introduction of the RNAs were transferred into the oviducts of surrogate female rats that had been anesthetized with isoflurane (DS Pharma Animal Health Co., Ltd., Japan).

## Genotyping analysis

Genomic DNA was extracted from the tail tips of 4-week-old rats using a KAPA Express Extract DNA Extraction Kit (Kapa Biosystems, London, UK). The genotyping primers for *Il2rg* were 5′-GACCAGAGGGGATTGCTGAG-3′ and 5′-GGTAGGTACCACATCTGCCC-3′; for *Rag2*, the genotyping primers were 5′-GGGGAGAAGGTGTCTTACGG-3′ and 5′-AGGTGGGAGGTAGCAGGAAT-3′. The PCR reaction mixture contained 200 µM dNTPs, 1.0 mM MgCl$_2$, and 0.66 µM of each primer in a total volume of 15 µl. The PCR cycles were as follows: one cycle at 94˚C for 3 min, followed by 35 cycles at 94˚C for 30 s, 60˚C for 1 min, and 72˚C for 45 s. The PCR products were directly sequenced with BigDye Terminator v3.1 Cycle Sequencing Mix on an Applied Biosystems 3130 DNA Sequencer (Life Technologies), in accordance with the manufacturer's standard procedure.

## Enzyme-linked immunosorbent assay (ELISA)

Whole blood samples were collected from the hearts of 18–21-week-old rats; for each sample, the serum was separated by centrifugation. The main serum immunoglobulin levels of rats with different genotypes were evaluated using rat ELISA quantitation kits for immunoglobulins IgG, IgM, and IgA (Bethyl Laboratories, Montgomery, TX, USA). The dilution ratio of serum is 1:50000 for IgG, 1:1000 for IgA, and 1:1000 for IgM. The concentrations of all immunoglobulins were calculated according to the absorbance values, which were evaluated using an iMark™ plate reader (cat. #168–1135, Bio-Rad, Hercules, CA, USA).

### RT-PCR

Total RNA was extracted from the spleens of 9-week-old WT F344/Jcl female, 16-week-old sKO female, and 14-week-old dKO female rats using a FastGene™ RNA Premium Kit (Nippon Genetics, Tokyo, Japan). First-strand cDNA was prepared from 1 µg of total RNA using Rever-Tra Ace® qPCR RT Master Mix (Toyobo, Osaka, Japan). The primers for *Il2rg* were 5′-CCG ACCAACCTCACTATGCA-3′ and 5′-GATTCTCTGGAGCCCATGGG-3′; for *Rag2*, the primers were 5′-AAGGCAGCACAGACTCTGAC-3′ and 5′-TCCTGGCAAGACAGTGCAAT-3′; and for *Gapdh*, the primers were 5′-GGCACAGTCAAGGCTGAGAATG-3′ and 5′-ATGGTG GTGAAGACGCCAGTA-3′. Assays were performed using KOD One® PCR Master Mix-Blue (Toyobo), as follows: 30 cycles at 98°C for 10 s, 60°C for 5 s, and 68°C for 3 s.

### Blood tests and flow cytometry

Hematological and biochemical parameters were assayed using a VetScan HM2 hematology system and VetScan VS2 (Zoetis, Parsippany, NJ, USA). For flow cytometry analysis of cell populations, peripheral blood was collected from the hearts of WT F344, sKO, and dKO rats. Specimens were lysed using ACK lysing buffer (Thermo Fisher Scientific), then analyzed using FITC anti-rat CD3 (clone 1F4), PE/Cy7 anti-rat CD4 (clone W3/25), APC anti-rat CD8a (clone G28), PE/Cy7 anti-rat CD45RA (clone OX33), and APC anti-rat CD161 (clone 3.2.3) (all antibodies from Biolegend, San Diego, CA, USA). Mouse IgG1 kappa, IgG2A kappa, and IgM kappa antibodies (Biolegend) were used as isotype controls. All cell samples were treated with mouse anti-rat CD32 (BD Biosciences, San Jose, CA, USA) for Fc receptor (FcR) blocking, then incubated with specific antibodies. The incubation time was 30 min at 4°C, and the assay was performed using a BD FACS Canto II cytometer (BD Biosciences).

### Statistical analysis

All statistical analyses were performed using R software, version 3.1.0 (https://www.r-project. org/) and GraphPad Prism, version 9.3.1 (GraphPad Software, San Diego, CA, USA). *p*-values <0.05 were considered statistically significant.

## Results

### Generation of *Il2rg* and *Rag2* knockouts using CRISPR/Cas9

We adopted the efficient rat genome editing strategy that was established in our previous study [10]. Two gRNAs targeting exon 2 of *Il2rg* or exon 3 of *Rag2* (Fig 1A) were introduced into 76 zygotes of F344/Jcl rats, together with Cas9 mRNA, by electroporation. After *in vitro* culture, 73 (96.1%) zygotes developed to the 2-cell stage and were transplanted into the oviducts of four surrogate female rats; 29 (39.7%) F0 offspring were obtained (Fig 1B). Targeted sequence analysis of the offspring revealed that six carried mutations in *Il2rg*, three carried mutations in *Rag2*, and two (No. 19 and No. 26) simultaneously carried mutations in both genes (Fig 1A). Only offspring No. 19 carried frame-shift mutations in both *Il2rg* (5-bp del) and *Rag2* ([1-bp del+2-bp ins]/[7-bp del+2-bp ins]). We crossed offspring No. 19 with a WT F344/Jcl male rat, then used the F1 generation to establish two immunodeficient rat models with genotypes $Il2rg^{-/-}(Il2rg^{-/Y})/Rag2^{-/-}$ and $Il2rg^{-/-}(Il2rg^{-/Y})$. The mutations in *Il2rg* and *Rag2* were subsequently confirmed by the lack of expression for the respective mRNAs (Fig 1C). In a previous study, we generated an immunodeficient rat model by simultaneous knockout of *Il2rg* and *Prkdc*, which encodes the DNA-activated protein kinase catalytic subunit. However, severe immunodeficiency in those rats was accompanied by growth retardation [19]. Therefore, to assess the health of dKO and sKO rats created in the present study, we

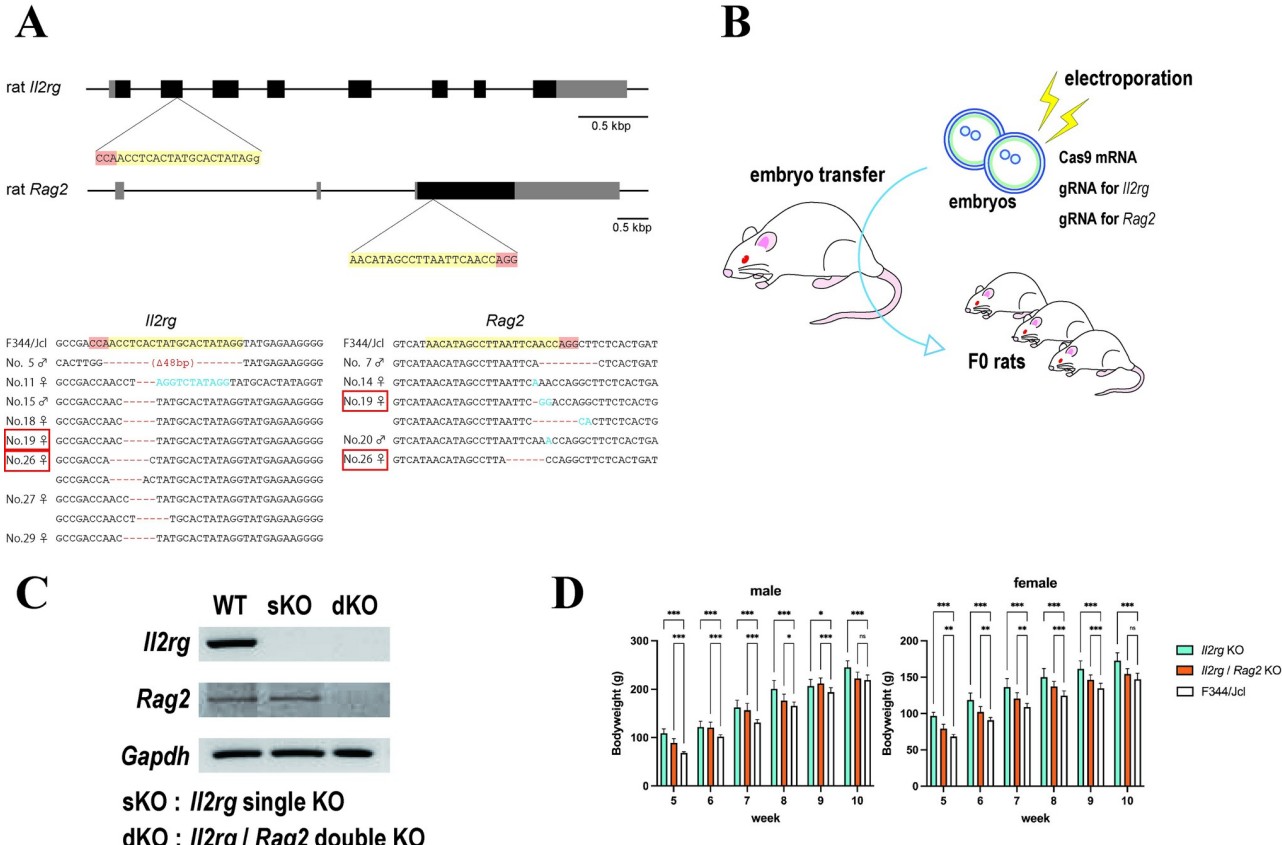

**Fig 1. Knockout of the *Il2rg* and *Rag2* genes.** (A) Schematic diagram of the gene knockout strategy and targeted sequences of the mutant F0 progenies. Blocks indicate exon regions. Guide RNAs (gRNAs) targeted to exon 2 of *Il2rg* and exon 3 of *Rag2* are highlighted in yellow; the protospacer adjacent motif (PAM) sequences are highlighted in pink. Targeted sequences of the F0 progenies are listed for comparison with the wild-type sequences. Short red lines indicate deletions and blue characters indicate insertions. Progenies that simultaneously carry mutations in both *Il2rg* and *Rag2* are indicated with red boxes. (B) Schematic representation of the method used for CRISPR-based knockout experiments in rat embryos. (C) RT-PCR analyses of *Il2rg* and *Rag2* from single-knockout (sKO) and double-knockout (dKO) rats (15-week-old) were conducted. *Gapdh* was used as the control. Here, images of different gels were partially cropped to facilitate display as a single figure. (D) Body weights of F344/Jcl wild-type, *Il2rg*-sKO, and *Il2rg/Rag2*-dKO rats were measured weekly, beginning at the fifth week postpartum. The graphs show the body weights of male (left) and female (right) rats. For both sKO and dKO rats, six male and six female rats were assessed. The reference weight data of the F344/Jcl wild-type rats were provided by CLEA Japan, Inc. (Tokyo, Japan) (n = 20). sKO and dKO rats were sometimes significantly heavier than the wild-type rats, but no growth retardation was observed. Data are presented as means with standard deviations. Multiple comparisons of each group (sKO or dKO) versus F344/Jcl were conducted using Dunnett's test. Asterisks or "ns" represent adjusted *p*-values: ns, 0.12; *, 0.033; **, 0.002; ***, <0.001.

evaluated their body weights beginning at the fifth week postpartum; we confirmed that their development was comparable to the development of WT F344/Jcl rats.

## The SCID rat models exhibit severe immunodeficiency phenotypes

Immunodeficiency was preliminarily confirmed by the presence of severe thymic hypoplasia in *Il2rg*-sKO and *Il2rg/Rag2*-dKO rats, whereas thymic morphology was similar between double heterozygous mutant female rats and WT female rats (Fig 2A). Additionally, spleen size was smaller in sKO female rats than in heterozygous mutant female rats (Fig 2B). Therefore, we presumed that normal immune function was maintained in heterozygous mutant rats. The anatomical phenotypes observed in our sKO and dKO rats are similar to the anatomical phenotypes of immunocompromised rats reported in previous studies [18]. Examination of blood cell counts in peripheral blood showed that the white blood cell count was significantly

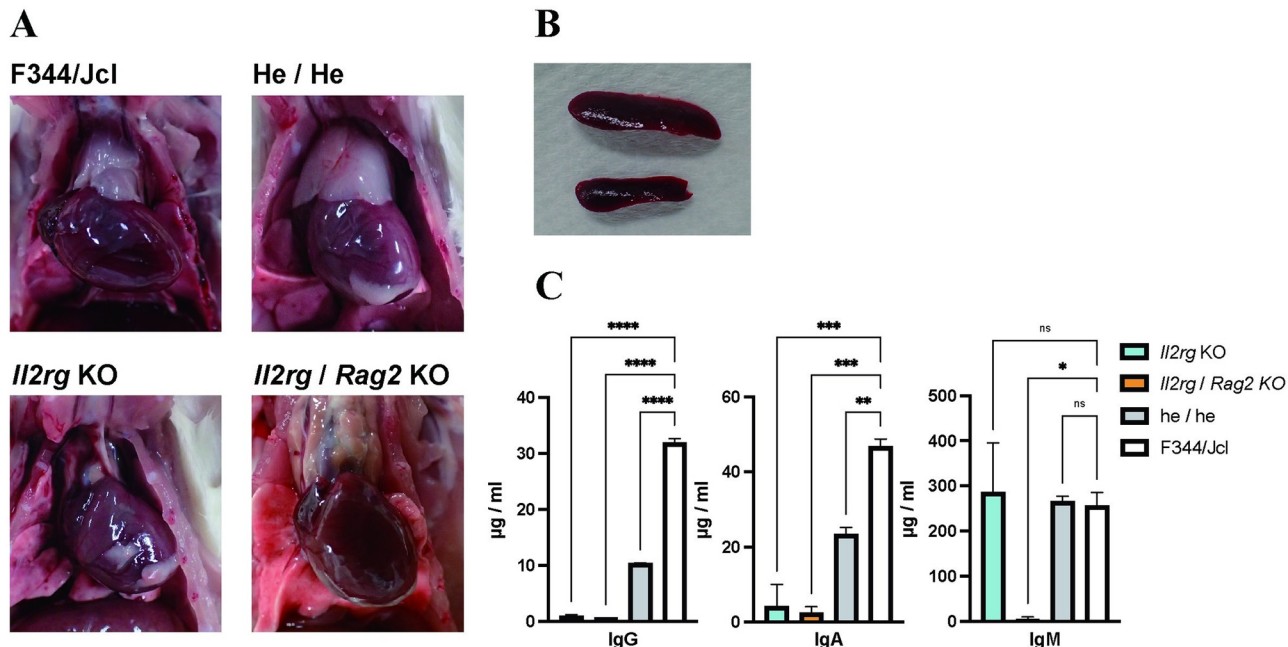

**Fig 2. Knockout of *Il2rg* and *Rag2* caused deficient development of lymphocytes.** (A) Images of thymus specimens from F344/Jcl wild-type, double heterozygote, *Il2rg*-single-knockout (sKO), and *Il2rg/Rag2*-double-knockout (dKO) rats. (B) Images of spleen specimens from 4-week-old double heterozygote (top) and *Il2rg*-sKO (bottom) rats. (C) Serum IgM, IgG, and IgA levels in F344/Jcl wild-type, double heterozygote, *Il2rg*-sKO, and *Il2rg/Rag2*-dKO rats, measured by ELISA. Multiple comparisons of each group (sKO, dKO, or he/he; double heterozygote) versus F344/Jcl were conducted using Dunnett's test. Asterisks or "ns" represent adjusted *p*-values: ns, 0.1234; *, 0.0332; **, 0.0021; ***, <0.0002; ****, <0.0001.

decreased in sKO and dKO rats, while the red blood cell count was unaffected (Fig 3A). The decrease in white blood cell count was mainly caused by a decrease in lymphocytes; the numbers of monocytes and granulocytes were similar among sKO, dKO, and WT rats (Fig 3B–3D). Analyses of lymphocyte populations via flow cytometry showed that T cells (SSC low/CD3$^+$) had been eliminated from dKO rats (Fig 4A). Furthermore, most T cells had been eliminated from *Il2rg* sKO rats: CD3$^+$ CD8$^+$ T cells were entirely absent, while some CD3$^+$ CD4$^+$ T cells remained (Fig 4A and 4B). The numbers of B cells (CD3$^-$ CD45$^+$) and NK cells (CD3$^-$ CD161$^+$) were also markedly reduced in sKO and dKO rats. Surprisingly, the low numbers of NK cells (CD3$^-$ CD161$^+$) were comparable between sKO and dKO rats (Fig 4C). ELISA analyses showed that serum levels of immunoglobulin IgG, IgA, and IgM were reduced in dKO rats, indicating a lack of adaptive immunity (Fig 2C). sKO rats had decreased serum levels of IgA and IgG, but they exhibited serum IgM levels similar to the levels in WT rats (Fig 2C). Heterozygous dKO rats also exhibited low levels of IgA and IgG, with IgM levels similar to the levels in WT rats; thus, we presume that a null mutation in *Rag2* is necessary for the loss of serum IgM. These findings indicated that both sKO and dKO rats exhibited immunodeficiency traits. sKO and dKO rats differed according to the presence or absence of IgM, as well as the small number of CD3$^+$ CD4$^+$ T cells that persisted in sKO rats.

## The SCID rat bioresource

As noted earlier in the manuscript, we previously generated an immunodeficient rat model by simultaneous knockout of *Il2rg* and *Prkdc*, which encodes the DNA-activated protein kinase catalytic subunit. However, the severe immunodeficiency of these rats was accompanied by growth retardation [19]. To assess the health of dKO and sKO rats created in the present

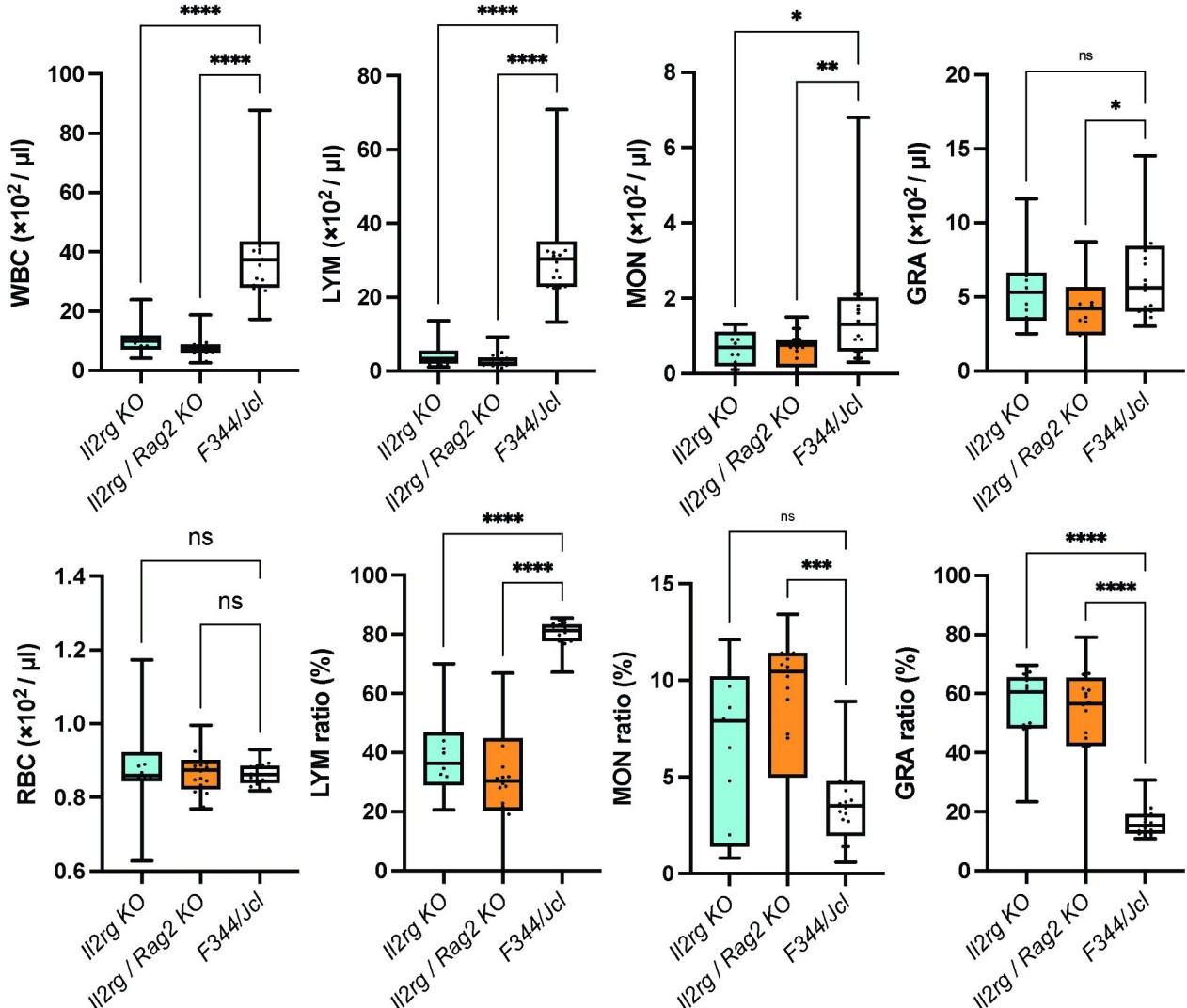

**Fig 3. Analysis of hematological parameters.** (A) Numbers of white blood cells (WBCs) and red blood cells (RBCs) in wild-type (WT), single-knockout (sKO), and double-knockout (dKO) rats. (B–D) Numbers of lymphocytes (LYM), monocytes (MON), granulocytes (GRA), and their ratios relative to all blood cells. The threshold of statistical significance was set at $p = 0.0167$ (Bonferroni correction of $p = 0.05/3$). WT, n = 20; sKO, n = 13; dKO, n = 20.

study, we evaluated their body weights beginning at the fifth week postpartum; we did not observe significant changes in growth speed among the dKO and sKO rats over 5 weeks of evaluation (Fig 1D). For use as a high-quality bioresource, the health condition and reproductive ability of SCID rats are critical considerations. The biochemical parameters were similar among dKO, sKO, and WT F344/Jcl rats (Table 1). Additionally, the mean offspring number in each litter indicated that large-scale breeding of both dKO (7.6 ± 3.096) and sKO (7.8 ± 2.927) rats would be successful. Using the F8 generation, we established a bioresource of dKO-SCID and sKO-SCID rats at The University of Tokyo in the National BioResource Project-Rat (SCID Rat by National BioResource Project-Rat [https://www.ims.u-tokyo.ac.jp/animal-genetics/scid/]). These high-quality SCID rats are bred in vinyl isolators and routinely examined for multiple microbiological infections (Table 2) at intervals of 1 or 3 months. The sperm and fertilized ova of the SCID rat models are also periodically collected and

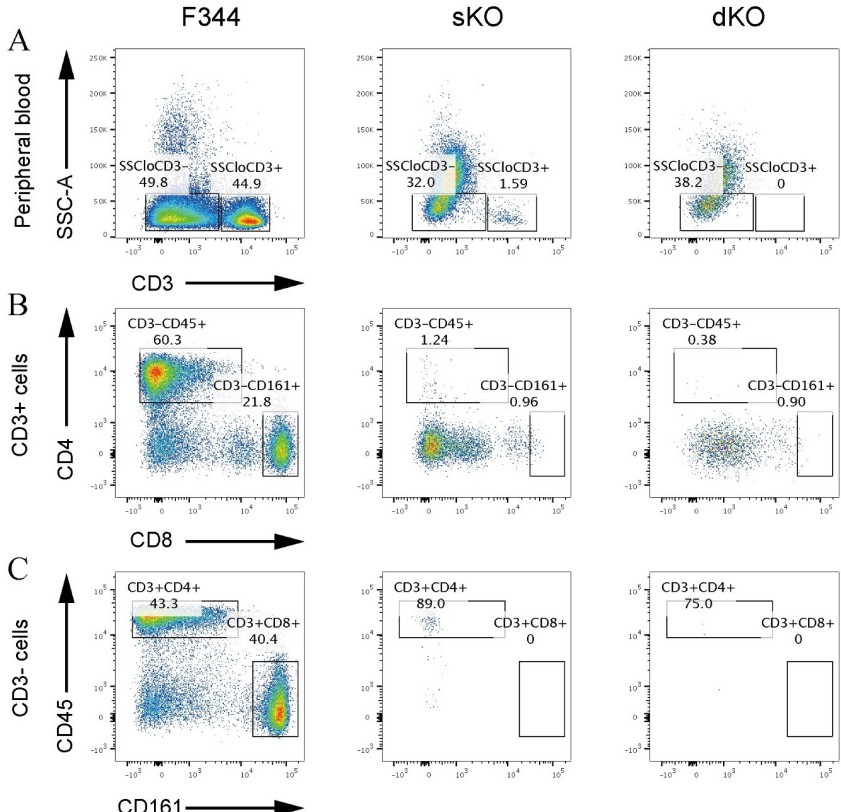

**Fig 4. Flow cytometry analysis of cell populations in peripheral blood.** (A) Analyses of SSC and CD3. Dot plots show the distributions of the cells after lymphoid gating. The SSC low/CD3$^+$ T cell group is indicated in the box as SSCloCD3$^+$. (B) Analyses of CD4 and CD8. Dot plots show the distributions of CD3$^+$ cells. (C) Analyses of CD45RA and CD161a. Cells from the SSCloCD3$^-$ gate were divided into groups. CD3$^-$/CD45RA$^+$ B cells and CD3$^-$/CD161a$^+$ NK cells are shown in the boxes.

cryopreserved as a reserve resource. Moreover, after superovulation treatment, comparable amounts of fertilized ova were obtained from SCID (i.e., dKO) and WT F344/Jcl rats (17.8 per rat from SCID rats [n = 18 rats]; 20.5 per rat from WT rats [n = 44 rats]). Our results indicate that these SCID rats are suitable for long-term maintenance breeding; they can be used in studies that require animals without growth retardation. Furthermore, the successful preservation of fertilized eggs and sperm indicates that these animals can serve as a stable bioresource.

## Discussion

Considering the limited availability of humanized rats and the benefits of experimentation involving laboratory rats (e.g., greater blood volume and ease of surgical procedures), we used the CRISPR/Cas9 system to generate SCID rats with mutations in *Il2rg* and *Rag2* (i.e., *Il2rg*-sKO and *Il2rg/Rag2*-dKO) that can serve as high-quality rat bioresources. These rats were generated using the genetically homogeneous F344/Jcl inbred line, which facilitated analyses of the phenotypic changes associated with experimental perturbations.

In the past few decades, humanized immunodeficient mice have been rapidly developed; they are useful for research in fields such as oncology, immunology, and immunotherapy [22, 23]. Previous studies have shown that approximately 30% of patients with SCID lack B cells; approximately half of these B cell-deficient patients with SCID have mutations in the *RAG*

**Table 1. Comparison of biochemical parameters among F344/Jcl, sKO, and dKO rats.**

| Parameters | Normal range | F344/Jcl (n = 12) | sKO (n = 12) | dKO (n = 20) |
|---|---|---|---|---|
| ALB (g/dL) | 2.1–4.6 | 5.73±0.22 | 5.63±0.33 | 5.56±0.26 |
| ALP (U/L) | 24–336 | 330.92±40.69 | 176.83±55.22*** | 183.05±65.86*** |
| ALT (U/L) | 51–138 | 56.42±9.79 | 39.50±8.27 | 48.85±28.19 |
| AMY (U/L) | 120–1436 | 473.58±37.70 | 581.08±107.75 | 599.80±128.86* |
| TBIL (mg/dL) | 0.1–0.5 | 0.27±0.089 | 0.25±0.052 | 0.24±0.050 |
| BUN (mg/dL) | 13–21 | 18.42±1.56 | 15.50±2.65*** | 15.15±1.23*** |
| CA (mg/dL) | 9.5–10.8 | 10.83±0.36 | 10.96±0.28 | 11.08±0.37 |
| PHOS (mg/dL) | 4.9–9.2 | 7.29±1.46 | 7.13±1.46 | 7.02±1.38 |
| CRE (mg/dL) | 0.3–0.4 | 0.39±0.14 | 0.56±0.16 | 0.42±0.20 |
| GLU (mg/dL) | 145–224 | 186.17±54.99 | 176.25±64.41 | 208.50±73.83 |
| NA+ (mmol/L) | 141–149 | 144.92±3.55 | 144.42±2.27 | 144.25±2.57 |
| K+ (mmol/L) | 3.7–4.7 | 5.50±1.11 | 5.79±0.68 | 5.78±1.09 |
| TP (g/dL) | 3.6–7.7 | 6.34±0.28 | 6.43±0.27 | 6.47±0.36 |
| GLOB (g/dL) | 0.4–3.5 | 0.60±0.15 | 0.83±0.33 | 0.93±0.36 |

Whole blood from adult rats was analyzed.

sKO, single-knockout; dKO, double-knockout. ALB, albumin; ALP, alkaline phosphatase; ALT, alanine aminotransferase; AMY, amylase; TBIL, total bilirubin; BUN, blood urea nitrogen; CA, calcium; PHOS, phosphorus; CRE, creatinine; GLU, glucose; NA+, sodium; K+, potassium; TP, total protein; GLOB: globulin.

*$p < 0.05$,

***$p < 0.01$ (Bonferroni correction).

genes. These findings highlighted the importance of the *RAG* genes in human SCID; researchers have since generated *Rag1* and *Rag2* sKO mice, as well as *Rag1/Rag2* dKO mice [13, 24, 26–28]. However, the loss of *Rag1* in mice has been shown to affect the central nervous system [29]; moreover, low expression levels of *Rag2* in mice only affected the development of T and B lymphocytes [13]. The *Il2rg* mutation differs from loss of function involving *Rag1* or *Rag2* in that it is characterized by the absence of NK cells and T cells. It has been reported that 46% of SCID patients in the United States have abnormal *IL2RG*. This high prevalence suggests that the *Il2rg* knockout may be an appropriate model for immunodeficiency. Additionally, *Il2rg* mutant mice lack B cells, which differs from the abnormal *IL2RG* phenotype in humans. *Il2rg* null mutant mice, which lack NK cells, have been used as a helpful SCID model because of their ability to accept engrafted human cells *in vivo* [16, 18, 25, 30].

In *Prkdc* SCID mice, spontaneous mutation of *Prkdc* disrupts the V(D)J recombination process in lymphocytes; it leads to the elimination of T and B cells [3, 31, 32]. To study various human tissues and cells *in vivo*, several immunodeficient mouse models based on *Prkdc* SCID have been created. Examples include the NOD.Cg-*Kit*$^{W-41J}$ *Prkdc*$^{scid}$ *Il2rg*$^{tm1Wjl/WaskJ}$ mouse (lacking T cells, B cells, NK cells, and hematopoietic stem cells) [33] and the NOD.Cg-*Prkdc*$^{scid}$ *Il2rg*$^{tm1Wjl}$ Tg(HLA-A/H2-D/B2M)1Dvs/SzJ mouse (lacking T cells, B cells, and NK cells; expressing human HLA class I heavy and light chains) [34].

In a previous study, we generated a null mutation in *Prkdc* via genome editing in a rat model [18]. We also observed significant growth retardation (70% body weight, compared with WT rats), although the SCID phenotype of the rat previous reported was similar [19]. In the present study, we found that T cells, B cells, and immunoglobulins were absent from *Il2rg/ Rag2*-dKO rats; moreover, NK cells were nearly absent from *Il2rg*-sKO and -dKO rats (a requirement for long-term humanization). The immunodeficiency phenotypes exhibited by *Il2rg*-sKO and *Il2rg/Rag2*-dKO rats were comparable to the phenotypes of our previously

**Table 2. Microbiological monitoring results.**

| Items | Methods | times/year |
|---|---|---|
| *Pasteurella pneumotropica* | Culture | 12 |
| *Bordetella bronchiseptica* | Culture | 12 |
| *Streptococcus pneumoniae* | Culture | 12 |
| *Corynebacterium kutscheri* | Culture | 12 |
| *Pseudomonas aeruginosa* | Culture | 12 |
| *Salmonella spp*. | Culture | 12 |
| *Staphylococcus aureus* | Culture | 12 |
| *Mycoplasma pulmonis* | Culture | 12 |
| *Corynebacterium kutscheri* | Serology | 12 |
| *Salmonella typhimurium* | Serology | 12 |
| *Clostridium piliforme* | Serology | 12 |
| *Mycoplasma pulmonis* | Serology | 12 |
| **Sialodacryoadenitis virus** | Serology | 12 |
| **Sendai virus** | Serology | 12 |
| **Mouse adenovirus (FL)** | Serology | 12 |
| *Giardia muris* | Microscopy | 12 |
| *Spironucleus muris* | Microscopy | 12 |
| *Syphacia spp*. | Microscopy | 12 |
| *Aspiculuris tetraptera* | Microscopy | 12 |
| **Hantavirus** | Serology | 4 |
| **LCM virus** | Serology | 4 |
| *Helicobacter bilis* | PCR | 4 |
| *Helicobacter hepaticus* | PCR | 4 |
| *Pneumocystis carinii* | PCR | 2 |
| **Ectoparasites** | Microscopy | 2 |
| **Intestinal protozoa** | Microscopy | 2 |
| **Pinworm** | Microscopy | 2 |
| **CAR bacillus** | Serology | 1 |
| **H-1 virus** | Serology | 1 |
| **Kilham rat virus** | Serology | 1 |
| **Mouse Minute virus** | Serology | 1 |
| **Mouse encephalomyelitis virus** | Serology | 1 |
| **Pneumonia virus of mice** | Serology | 1 |
| **Reovirus type 3** | Serology | 1 |
| **Rat Polyoma virus 2** | PCR | 1 |

developed immunodeficient rat models [18, 19]. Previous models have been used in the investigation of intestinal immunity to parasites [35], the creation of a humanized rat model of osteosarcoma [36], and the regeneration of rat laryngeal cartilage by human induced pluripotent stem cell-derived mesenchymal stem cells [37]. These previous studies clearly demonstrate the usefulness of immunodeficient rats. However, conventional immunodeficient rats exhibit growth retardation as mentioned above; in these rats, contamination with rat polyoma virus 2 is a concern because of their increased susceptibility to infection [38]. Importantly, our SCID rats exhibit normal growth; thus far, they have been free of rat polyoma virus 2 infection (Table 2).

This study had some limitations. First, it did not confirm graft viability. Future studies of these rats should involve transplantation experiments with human cancer cell lines, other strains of rat skin grafts, and human hematopoietic stem cells. Second, the hematological analysis was limited; more detailed hematological investigation is needed in a future study. Third, the ELISA analysis included a limited number of repeated tests for each sample (n = 2), which may have led to low detection power. Additional replicates should be included in future studies.

Several immunodeficient rat models have been generated by single or combined knockouts of *Prkdc*, *Rag1*, *Rag2*, and *Il2rg* [1, 15, 19, 39]; most of these models were established in outbred rat lines. If a model is mainly intended to serve as a recipient for the transplantation of exogenous cells, tissues, or medical materials, the relatively large body-sized outbred strains may be helpful. However, inbred rats may be more appropriate if future plans include comparisons involving some form of treatment or genetic perturbation. To our knowledge, our model is the first high-quality SCID rat model to be created by knocking out *Rag2* and/or *Il2rg* among inbred rats with the F344/Jcl genetic background. Using these high-quality SCID rats, we constructed a SCID rat bioresource in the National BioResource Project-Rat at the University of Tokyo (https://www.ims.u-tokyo.ac.jp/animal-genetics/scid/) [21]. Our SCID rats have demonstrated stability during experimental studies by research institutes and researchers worldwide. Furthermore, various genome modification and utilization studies are ongoing. Our SCID rat model may provide a foundation for practical applications of personalized medicine in cancer treatment; it may also be useful in preclinical research regarding bone regeneration, which would benefit from the larger body size in rats (compared to mice) [40–42]. In conclusion, our novel SCID rats are expected to be useful in a wide range of applications that extend beyond transplantation studies.

The S1 Fig file is the raw gel images of Fig 1B; the squared areas of A and B were cropped and combined to form Fig 1B.

A was not processed. B is an image with automatic contrast adjustment. C is the original image of B.

## Supporting information

**S1 Fig. Original images for representative RT-PCR gels shown in Fig 1C.** Three animals per group were randomly selected for RT-PCR analysis. DNA samples were separated by electrophoresis in 1.5% agarose gel; the gels were stained with Invitrogen™ SYBR™ Safe DNA Gel Stain (Thermo Fisher), imaged under the blue/green LED light (Blue/Green LED transilluminator, Nippon Genetics CO., LTD. Tokyo, Japan) by a gel imager (Gel Scene GST-33, Astec CO., Ltd. Fukuoka, Japan). Image A was analyzed without adjustment. Image B (used for analysis) is an automatic contrast-adjusted version of image C; image C is the unaltered version of image B. Lanes 1, 5, 9, 13 and 24 contain DNA size markers. Lane No. 23 is a blank lane. Rectangular outline areas in images A and B were cropped (*Il2rg*: Lanes 2, 3, 4; *Gapdh*: Lanes 10, 11, 12 in image A. *Rag2*: Lanes 14, 15, 16 in image B) and combined to form Fig 1C. *Rag2* electrophoresis was performed using distinct primers for image A vs. images B and C. Lanes 14, 15, and 16, which demonstrated specific amplification, are shown in Fig 1C.
(PDF)

## Acknowledgments

We thank Kenji Toyonaga, PhD, for FACS data analysis and Kanako Shimizu, MSc, for cryopreservation of the sKO and dKO embryos. We are thankful for the grant to the National

BioResource Project-Rat (http://www.anim.med.kyoto-u.ac.jp/NBR/). We thank Margaret Biswas, PhD, and Ryan Chastain-Gross, PhD, from Edanz (https://jp.edanz.com/ac) for editing a draft of this manuscript.

## Author Contributions

**Conceptualization:** Tomoji Mashimo.

**Data curation:** Yoshiki Miyasaka.

**Formal analysis:** Yoshiki Miyasaka.

**Funding acquisition:** Tomoji Mashimo.

**Investigation:** Yoshiki Miyasaka, Jinxi Wang, Kosuke Hattori, Yuko Yamauchi.

**Project administration:** Tomoji Mashimo.

**Supervision:** Tomoji Mashimo.

**Visualization:** Yoshiki Miyasaka.

**Writing – original draft:** Yoshiki Miyasaka, Jinxi Wang.

**Writing – review & editing:** Kosuke Hattori, Yuko Yamauchi, Miho Hoshi, Kazuto Yoshimi, Saeko Ishida, Tomoji Mashimo.

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
