## [Decision Letter · Decision Letter 0]

29 Sep 2021

PONE-D-21-27266Development of a high-quality bio-resource of severe combined immunodeficient (SCID) rat models with Il2rg and/or Rag2 mutationsPLOS ONE

Dear Dr. Mashimo,

Thank you for submitting your manuscript to PLOS ONE. After careful consideration, we feel that it has merit but does not fully meet PLOS ONE’s publication criteria as it currently stands. Therefore, we invite you to submit a revised version of the manuscript that addresses the points raised during the review process.

We look forward to receiving your revised manuscript.

Kind regards,

Bing He

Academic Editor

PLOS ONE

Journal Requirements:

2. In your Methods section, please include a comment about the state of the animals following this research. Were they euthanized or housed for use in further research? If any animals were sacrificed by the authors, please include the method of euthanasia and describe any efforts that were undertaken to reduce animal suffering.

Reviewers' comments:

Reviewer's Responses to Questions

**Comments to the Author**

1. Is the manuscript technically sound, and do the data support the conclusions?

Reviewer #1: Partly

Reviewer #2: Yes

2. Has the statistical analysis been performed appropriately and rigorously? 

Reviewer #1: Yes

Reviewer #2: Yes

3. Have the authors made all data underlying the findings in their manuscript fully available?

Reviewer #1: No

Reviewer #2: Yes

4. Is the manuscript presented in an intelligible fashion and written in standard English?

Reviewer #1: Yes

Reviewer #2: Yes

5. Review Comments to the Author

Reviewer #1: - More suitable title should be selected for the article. Title should decrease to 10-12 words.

- The abstract should state briefly the purpose of the research, the principal results and major conclusions. An abstract is often presented separately from the article, so it must be able to stand alone.

- It is suggested to present the structure of the article at the end of the introduction.

- The major defect of this study is the debate or Argument is not clear stated in the introduction session. Hence, the contribution is weak in this manuscript. I would suggest the author to enhance your theoretical discussion and arrives your debate or argument.

- More suitable title should be selected for the table 1 instead of “Blood biochemistry results.”.

- Page 6: the following paragraph is unclear, so please reorganize that:

“The Osaka University Animal Experiment Committee approved all the animal experiments (Permission number:24-006-042). The Il2rg/Rag dKO and Il2rg sKO-SCID rats were microbiologically tested by the Fujinomiya Technical Service Center (FTSC) of CLEA Japan, where they were also kept in an IVC system.”

- It is suggested to add articles entitled “Ehnert et al. Feasibility of Cell Lines for In Vitro Co-Cultures Models for Bone Metabolism”, “Kosvyra et al. Developing an Integrated Genomic Profile for Cancer Patients with the Use of NGS Data” and “Abdul Abubakar et al. Generation of Open Metatarsal Fracture in Rats: A Model for Secondary Fracture Healing” to the literature review.

- Much more explanations and interpretations must be added for the Results, which are not enough.

- Please make sure your conclusions' section underscore the scientific value added of your paper, and/or the applicability of your findings/results, as indicated previously. Please revise your conclusion part into more details. Basically, you should enhance your contributions, limitations, underscore the scientific value added of your paper, and/or the applicability of your findings/results and future study in this session.

- It is suggested to compare the results of the present research with some similar studies which is done before.

- DOI of the references must be added (you can use “" ext-link-type="uri" xlink:type="simple">https://crossref.org/").

Reviewer #2: This study successfully established a severe immunodeficiency rat model. This model showed significantly reduced immune cells without significant growth retardation or defective gametogenesis. It is undoubtedly of positive significance for humanized studies. Comments are the following:

1. Line 164 Page 11, for the generation of SCID rat, it is better to present a schematic diagram of workflow.

2. Line 155 Page 8, The list of tumors seems redundant. It is more appropriate to show it in the table. In addition, the abbreviation for the tumor name is repeated in the figure legend (Figure 2 3).

3. In Figure 5, left panel and right panel need to be labeled with gender information.

4. Please label the statistical comparison results in figures (Fig. 2c and Figure 5) and indicate the statistical method used in the legend.

5. The writing and the resolution of the figures in this manuscript could to be improved.

6. PLOS authors have the option to publish the peer review history of their article (what does this mean?). If published, this will include your full peer review and any attached files.

Reviewer #1: No

Reviewer #2: No

---

## [Author Response · Author response to Decision Letter 0]

18 Mar 2022

Responses to the comments of Reviewer #1

1. More suitable title should be selected for the article. Title should decrease to 10-12 words.

Response: In accordance with the reviewer’s comment, we have revised the manuscript title as follows:

“A high-quality severe combined immunodeficiency (SCID) rat bioresource”

2. The abstract should state briefly the purpose of the research, the principal results and major conclusions. An abstract is often presented separately from the article, so it must be able to stand alone.

Response: We have revised the abstract in accordance with the reviewer’s comment.

3. It is suggested to present the structure of the article at the end of the introduction.

Response: We have revised the Introduction in accordance with the reviewer’s comment.

4. The major defect of this study is the debate or Argument is not clear stated in the introduction session. Hence, the contribution is weak in this manuscript. I would suggest the author to enhance your theoretical discussion and arrives your debate or argument.

Response: We have revised the entire introduction in accordance with this comment and comment #3 by this reviewer.

5. More suitable title should be selected for the table 1 instead of “Blood biochemistry results.”.

Response: In accordance with the reviewer’s comment, we have revised the title of Table 1 as follows:

“Comparison of Biochemical Parameters Among F344/Jcl, sKO, and dKO Rats.”

6. Page 6: the following paragraph is unclear, so please reorganize that:

“The Osaka University Animal Experiment Committee approved all the animal experiments (Permission number:24-006-042). The Il2rg/Rag dKO and Il2rg sKO-SCID rats were microbiologically tested by the Fujinomiya Technical Service Center (FTSC) of CLEA Japan, where they were also kept in an IVC system.”

Response: We have revised the indicated text for clarity, in accordance with the reviewer’s comment.

“All rats were housed in an individually ventilated cage system; they received a standard diet and tap water ad libitum. Microbiological analyses of Il2rg/Rag dKO and Il2rg sKO-SCID rats were conducted by the Fujinomiya Technical Service Center of CLEA Japan. All animal experiments were approved by the Osaka University Animal Experiment Committee (approval number: 24-006-042).”

7. It is suggested to add articles entitled “Ehnert et al. Feasibility of Cell Lines for In Vitro Co-Cultures Models for Bone Metabolism”, “Kosvyra et al. Developing an Integrated Genomic Profile for Cancer Patients with the Use of NGS Data” and “Abdul Abubakar et al. Generation of Open Metatarsal Fracture in Rats: A Model for Secondary Fracture Healing” to the literature review.

Response: We thank the reviewer for these suggestions. We have cited them (Refs. 39, 40, and 41) in the context of potential research applications for immunocompromised rats.

8. Much more explanations and interpretations must be added for the Results, which are not enough.

Response: We have revised the Results in accordance with the reviewer’s comment.

9. Please make sure your conclusions' section underscore the scientific value added of your paper, and/or the applicability of your findings/results, as indicated previously. Please revise your conclusion part into more details. Basically, you should enhance your contributions, limitations, underscore the scientific value added of your paper, and/or the applicability of your findings/results and future study in this session.

Response: We presume that the reviewer is referring to the Discussion section because the manuscript does not contain a Conclusions heading. We have revised the Discussion in accordance with the reviewer’s comment.

10. It is suggested to compare the results of the present research with some similar studies which is done before.

Response: In accordance with the reviewer’s comment, we have revised the Introduction and Discussion to address similarities and differences of our immunodeficient rats with respect to existing animal models of immunodeficiency.

11. DOI of the references must be added (you can use “https://crossref.org/")..

Response: We have added DOI numbers for all references, in accordance with the reviewer’s comment.

Responses to the comments of Reviewer #2

1. Line 164 Page 11, for the generation of SCID rat, it is better to present a schematic diagram of workflow.

Response: In accordance with the reviewer’s comment, we have added the schematic diagram as Figure 1B.

2. Line 155 Page 8, The list of tumors seems redundant. It is more appropriate to show it in the table. In addition, the abbreviation for the tumor name is repeated in the figure legend (Figure 2 3).

Response: We have carefully checked the manuscript and could not find a list of tumors in the main text, or an abbreviated tumor name in the legends for Figures 2 and 3. We request additional clarification from the reviewer.

3. In Figure 5, left panel and right panel need to be labeled with gender information.

Response: In accordance with the reviewer’s comment, we have added the animal sex information in Figure 5 (Figure 1D in the revised manuscript).

4. Please label the statistical comparison results in figures (Fig. 2c and Figure 5) and indicate the statistical method used in the legend.

Response: In accordance with the reviewer’s comment, we have added the results of statistical comparisons to Figure 2C and Figure 5 (Figure 1D in the revised manuscript). We have also described the statistical comparison method in the legends and the statistical software in the Materials and Methods section.

5. The writing and the resolution of the figures in this manuscript could to be improved.

Response: In accordance with the reviewer’s comment, the manuscript has been carefully reviewed by an experienced editor whose first language is English and who specializes in editing papers written by researchers whose native language is not English. Additionally, we have improved the resolution for all figures.

---

## [Decision Letter · Decision Letter 1]

29 Jul 2022

A high-quality severe combined immunodeficiency (SCID) rat bioresource

PONE-D-21-27266R1

Dear Dr. Mashimo,

We’re pleased to inform you that your manuscript has been judged scientifically suitable for publication and will be formally accepted for publication once it meets all outstanding technical requirements.

Kind regards,

Bing He

Academic Editor

PLOS ONE

Additional Editor Comments (optional):

Reviewers' comments:

Reviewer's Responses to Questions

**Comments to the Author**

1. If the authors have adequately addressed your comments raised in a previous round of review and you feel that this manuscript is now acceptable for publication, you may indicate that here to bypass the “Comments to the Author” section, enter your conflict of interest statement in the “Confidential to Editor” section, and submit your "Accept" recommendation.

Reviewer #1: All comments have been addressed

2. Is the manuscript technically sound, and do the data support the conclusions?

Reviewer #1: Yes

3. Has the statistical analysis been performed appropriately and rigorously? 

Reviewer #1: Yes

4. Have the authors made all data underlying the findings in their manuscript fully available?

Reviewer #1: Yes

5. Is the manuscript presented in an intelligible fashion and written in standard English?

Reviewer #1: Yes

6. Review Comments to the Author

Reviewer #1: The authors have successfully addressed all my concerns in the revised manuscript. Hence I recommend the acceptance of this paper.

7. PLOS authors have the option to publish the peer review history of their article (what does this mean?). If published, this will include your full peer review and any attached files.

Reviewer #1: No

---

## [Editor Report · Acceptance letter]

4 Aug 2022

PONE-D-21-27266R1 

A high-quality severe combined immunodeficiency (SCID) rat bioresource 

Dear Dr. Mashimo:

I'm pleased to inform you that your manuscript has been deemed suitable for publication in PLOS ONE. Congratulations! Your manuscript is now with our production department. 

Kind regards, 

on behalf of

Dr. Bing He 

Academic Editor

PLOS ONE